# Anesthetic Management of Acute Ischemic Stroke Undergoing Mechanical Thrombectomy: An Overview

**DOI:** 10.3390/diagnostics14192113

**Published:** 2024-09-24

**Authors:** Alessandro De Cassai, Nicolò Sella, Tommaso Pettenuzzo, Annalisa Boscolo, Veronica Busetto, Burhan Dost, Serkan Tulgar, Giacomo Cester, Nicola Scotti, Alessandro di Paola, Paolo Navalesi, Marina Munari

**Affiliations:** 1Sant’Antonio Anesthesia and Intensive Care Unit, University Hospital of Padua, 35128 Padua, Italy; marina.munari@aopd.veneto.it; 2UOC Anesthesia and Intensive Care Unit, University Hospital of Padua, 35128 Padua, Italy; nicolo.sella@aopd.veneto.it (N.S.); tommaso.pettenuzzo@aopd.veneto.it (T.P.); annalisa.boscolobozza@unipd.it (A.B.); paolo.navalesi@unipd.it (P.N.); 3Department of Medicine—DIMED, University of Padova, 35131 Padova, Italy; 4Thoracic Surgery and Lung Transplant Unit, Department of Cardiac, Thoracic, Vascular Sciences, and Public Health, University of Padua, 35122 Padova, Italy; 5Cardiac Surgery Intensive Care Unit, University Hospital of Padua, 35128 Padua, Italy; veronica.busetto@aopd.veneto.it; 6Department of Anesthesiology and Reanimation, Ondokuz Mayis University Faculty of Medicine, Samsun 55220, Türkiye; burhan.dost@omu.edu.tr; 7Department of Anesthesiology and Reanimation, Samsun Training and Research Hospital, Samsun University Faculty of Medicine, Samsun 55280, Türkiye; serkantulgar.md@gmail.com; 8Department of Neruoradiology, University Hospital of Padua, 35128 Padua, Italy; giacomo.cester@aopd.veneto.it (G.C.); nicola.scotti@aopd.veneto.it (N.S.); alessandro.dipaola@aopd.veneto.it (A.d.P.)

**Keywords:** anesthesia, acute stroke, airway management, hemodynamic, hospital rapid response team

## Abstract

Ischemic stroke, caused by the interruption of the blood supply to the brain, requires prompt medical intervention to prevent irreversible damage. Anesthetic management is pivotal during surgical treatments like mechanical thrombectomy, where precise strategies ensure patient safety and procedural success. This narrative review highlights key aspects of anesthetic management in ischemic stroke, focusing on preoperative evaluation, anesthetic choices, and intraoperative care. A rapid yet thorough preoperative assessment is crucial, prioritizing essential diagnostic tests and cardiovascular evaluations to determine patient frailty and potential complications. The decision between general anesthesia (GA) and conscious sedation (CS) remains debated, with GA offering better procedural conditions and CS enabling continuous neurological assessment. The selection of anesthetic agents—such as propofol, sevoflurane, midazolam, fentanyl, remifentanil, and dexmedetomidine—depends on local protocols and expertise balancing neuroprotection, hemodynamic stability, and rapid postoperative recovery. Effective blood pressure management, tailored airway strategies, and vigilant postoperative monitoring are essential to optimize outcomes. This review underscores the importance of coordinated care, incorporating multimodal monitoring and maintaining neuroprotection throughout the perioperative period.

## 1. Introduction

Ischemic stroke originates from the interruption of blood supply to a part of the brain [1]. This interruption, often due to a blood clot or other blockage in a cerebral artery, deprives brain tissue of oxygen and nutrients, leading to cell death and potential loss of function in affected areas. However, ischemia of the brain tissue represents only the final step in the progression of stroke pathology and the patients have to go through several other reversible steps before that. For this reason, for each occurrence of ischemic stroke, the medical team is in a race against time to help move patients away from the brink of severe and irreversible damage; consequently, prompt intervention is crucial during stroke pathology to halt the progression and improve patient outcomes [2].

Anesthetic management plays a crucial role in the surgical treatment of ischemic stroke [3], particularly during procedures like mechanical thrombectomy.

Mechanical thrombectomy for stroke patients is a highly specialized, minimally invasive procedure aimed at swiftly restoring blood flow by removing the blood clot from the cerebral vessel [4,5]. To achieve successful reperfusion, the patient is placed under conscious sedation or general anesthesia; a neurointerventionalist then inserts a catheter into the femoral artery located in the groin and navigates it through the vascular system to the site of the clot in the brain, guided by real time usually by using fluoroscopy.

Once the catheter reaches the clot, various devices can be employed to remove it. Stent retrievers, which are mesh-like devices, can ensnare the clot, enabling it to be pulled out. Alternatively, aspiration catheters use suction to aspirate the clot directly from the vessel. After successfully removing the clot, blood flow is restored to the affected region of the brain, which is confirmed through angiography.

To guarantee the best outcome for the patient suffering from stroke, the anesthesiologist must consider multiple factors when approaching a patient affected by ischemic stroke. Given the complexity of this issue, our narrative review aims to provide practitioners with a comprehensive overview of anesthetic management in acute ischemic stroke. To achieve this, we will briefly explore preoperative evaluation, the choice between general anesthesia and conscious sedation, the selection of anesthetic agents, and intraoperative care strategies.

## 2. Preoperative Evaluation—Time Is Brain

Mechanical thrombectomy for patients affected by ischemic stroke is an emergency, leaving limited time for an extended evaluation of the patient. Nevertheless, this urgency necessitates an even more careful preoperative evaluation to assess the patient’s frailty, potential perioperative complications, and overall clinical status. Medications that the patient is currently taking, particularly anticoagulants and antiplatelet agents, require careful consideration in this context. Given that “time is brain” (Figure 1) [6,7], ventilator checks, drug preparation, and operating room checklists should be completed as soon as the stroke team is alerted, while patient assessment must be conducted in a as short a time as possible concurrently with patient monitoring.

In this framework, coordination and communication among the stroke team are paramount to streamline the preoperative process. The anesthesiologist is not the sole actor in this scenario; while they evaluate and assess the patient, other team members should prepare the necessary equipment and medications, establish intravenous access, initiate appropriate monitoring (including electrocardiogram [ECG], blood pressure, and oxygen saturation), prepare for potential airway management, and ensure that the operating room is ready without delay. These are essential steps for optimizing patient outcomes.

A thorough yet rapid assessment is crucial to determine the patient’s neurological baseline status, cardiovascular stability, and any comorbid conditions that might impact the procedure. This involves a quick but comprehensive review of the patient’s medical history, a focused physical examination, and essential diagnostic tests. The evaluation should prioritize identifying immediate risks, such as unstable cardiac conditions, severe hypertension, or significant coagulopathies that require prompt management. Different scales are usually used to quickly assess the patient’s neurological and functional status and the ability to tolerate anesthesia and surgery; for the former, usually the National Institutes of Health Stroke Scale (NIHSS) [8,9] is used, while for the latter, the American Society of Anesthesiologists (ASA) physical status classification [10,11] is used.

The National Institutes of Health Stroke Scale is a clinical tool used to evaluate the severity of a stroke by assessing 11 functions such as consciousness, vision, movement, sensation, speech, and language. Each item is scored from 0 (normal) to 4 (severe impairment), with a maximum score of 42, where higher scores indicate more severe strokes.

The ASA Physical Status Classification System (ASA-PS) is used by anesthesiologists to assess and communicate a patient’s pre-anesthesia medical comorbidities. It ranges from 1 to 6: 1 denotes a healthy patient, 2 a patient with mild systemic disease, 3 a patient with severe systemic disease, 4 a patient with severe systemic disease that is a constant threat to life, 5 a moribund patient not expected to survive without surgery, and 6 a declared brain-dead patient whose organs are being removed for donor purposes.

Finally, consultation with a neurologist (and in rare cases with other specialists) is necessary to assess the patient’s condition before surgery.

## 3. Preoperative Evaluation—Which Exams Should I Look for?

To keep it simple, no exams other than imaging studies, such as computed tomography (CT) or magnetic resonance imaging (MRI), which are essential for identifying the location and size of the infarct and ruling out hemorrhagic stroke, are mandatory when evaluating a patient with ischemic stroke [12]. As stated above, the priority of the anesthesiologists is to reduce the compressible time as much as possible in order to reduce the final damage the patient will suffer; however, during this brief window, the anesthesiologist should also assess tests that could offer crucial insights into the patient’s condition. First and foremost, obtaining results from essential blood tests is highly beneficial. A complete blood count, coagulation profile, and blood glucose levels are crucial. The coagulation profile, in particular, is critical due to the delicate balance between preventing thromboembolism and managing the risk of surgical bleeding.

Additionally, cardiovascular evaluation plays a significant role, given that many ischemic stroke patients are elderly and have underlying cardiovascular conditions. An ECG can provide a foundational assessment of cardiac function, helping to identify arrhythmias such as atrial fibrillation, which may have contributed to the stroke and may affect the anesthetic management. If available, a recent echocardiogram can offer further insight into cardiac function, revealing areas of previous myocardial infarction or valve disease, and guiding perioperative care. Although they are not directly related to patients suffering from ischemic stroke and cannot be directly applied due to the urgency of endovascular intervention, guidelines on preoperative risk assessment can still be useful for readers to understand the rationale behind each exam [13,14].

The importance of point-of-care ultrasound (POCUS) in the preoperative evaluation of stroke patients cannot be overstated. POCUS could provide crucial information to the anesthesiologist in real time with a minimum loss of time. Especially crucial is the cardiac examination for assessing hemodynamic status, which provides vital information about the patient’s cardiovascular condition. Additionally, a gastric examination using POCUS can detect potential regurgitation risks in unconscious patients, thereby aiding in securing the airway. These evaluations are indispensable for anesthesiologists as they navigate the complexities of managing emergency situations, ensuring a safer and more effective anesthetic plan. However, to ensure that POCUS is effective and not time-consuming, operators need proper training; moreover, an intrinsic limit of this type of imaging is that it is operator-dependent [15].

In summary, while immediate action is paramount in stroke emergencies, having access to these diagnostic results can enhance the anesthesiologist’s ability to manage the patient effectively. The goal is to integrate these evaluations efficiently into the urgent care process, balancing the need for rapid intervention with a thorough understanding of the patient’s health status.

## 4. Preoperative Evaluation—Not Only a Brain

The anesthesiologist should maintain open communication with the patient’s relatives, providing updates on the patient’s condition, the planned procedure, and any potential risks involved. This transparency helps alleviate the family’s anxiety and depression, common feelings in relatives of critical care patients [16]. Moreover, this approach helps foster trust in the overall medical staff. Additionally, the anesthesiologist should try to communicate directly with the patient, explaining the procedure and what to expect, tailored to the patient’s level of understanding. This approach acknowledges that the patient is not just a clinical case but a human being with concerns and emotions, thereby ensuring compassionate and patient-centered care. Ensuring that the patient feels informed and supported can significantly reduce their stress and improve cooperation, which is essential for a successful outcome. The anesthesiologist’s empathetic communication underscores the holistic nature of medical care, recognizing the patient’s dignity and humanity throughout the treatment process [17].

## 5. Patient Monitoring

Patients should undergo standard monitoring of vital parameters, including ventilation (end-tidal CO_2_), oxygenation (pulse oximetry), temperature, and circulation (electrocardiogram and blood pressure). Invasive blood pressure monitoring via an arterial catheter is preferred and ideally should be established before arrival at the neuro radiology suite, and this procedure can be performed under local anesthesia. However, unless specifically indicated, the procedure should not be delayed solely for arterial catheter placement. Noninvasive blood pressure monitoring with frequent cycling usually suffices for hemodynamic monitoring [18].

## 6. General Anesthesia or Conscious Sedation?

Choosing between general anesthesia (GA) and conscious sedation (CS) for patients suffering from ischemic stroke remains a highly debated topic in neuroanesthesia, with no clear consensus on the best approach. Current guidelines do not offer specific recommendations, instead suggesting that the decision should be individualized based on the patient’s characteristics [19]. Recent systematic reviews and meta-analyses indicate that the modified Rankin Scale (mRS) score at three months post-endovascular therapy is not significantly influenced by the chosen anesthetic strategy. Nevertheless, successful reperfusion rates appear higher in patients managed with general anesthesia (GA), although this does not translate into better functional outcomes [20,21].

Given this context, it is evident that the anesthesiologist managing such a frail patient must rely on their own experience, institutional protocols, and the patient’s specific conditions to make a decision; however, each approach has both advantages and disadvantages that we will now briefly discuss.

Proponents of GA argue that it provides better procedural conditions by ensuring immobility, controlling ventilation, and optimizing cerebral oxygenation and perfusion. GA can be particularly beneficial for patients who are agitated, have a predicted difficult airway, poor baseline oxygen saturation, dysphagia, or an inability to follow commands. Furthermore, GA may reduce the risk of patient movement during critical parts of the procedure, potentially increasing the chances of successful reperfusion. Studies have shown that GA might be associated with a higher rate of successful reperfusion, which is a key determinant of favorable outcomes [20,21]. However, the time from groin puncture to reperfusion is slightly longer in GA, possibly due to the time required for induction of anesthesia [20,22]. On the other hand, CS has the advantage of avoiding the risks associated with intubation and mechanical ventilation, such as aspiration, pneumonia, and hemodynamic instability. It allows for continuous neurological assessment during the procedure, which can be critical in adjusting the intervention in real-time based on the patient’s responses. An additional potential disadvantage of CS is the risk of emergent conversion to GA. This can delay stroke treatment, increase the risk of aspiration, and necessitate securing the airway in a suboptimal situation. Identifying predictive factors for emergent conversion such as tandem occlusion, prolonged endovascular procedures, patient comorbidities, and the number of pharmacological agents used during the procedure [23] could guide the anesthesiologist in opting for GA over CS initially.

Additionally, the identification of risk factors for emergent conversion to GA could influence the initial anesthetic strategy, potentially improving overall outcomes. The balance between optimizing procedural conditions and minimizing time to reperfusion remains a key consideration in choosing the most appropriate anesthetic approach for endovascular therapy in ischemic stroke patients.

## 7. Choosing the Right Drugs

When performing ischemic stroke thrombectomy, selecting the optimal anesthetic approach is crucial to ensure patient safety and procedural success. The choice of anesthetic agents in ischemic stroke patients involves balancing neuroprotection, hemodynamic stability, and rapid recovery for postoperative neurological assessment. There is no one-size-fits-all solution; the choice of anesthetic should be individualized based on the patient’s overall health status, hemodynamic condition, complexity of the thrombectomy procedure, and expertise of the medical team. Tailoring the anesthetic approach to these factors ensures optimal procedural conditions while effectively managing the patient’s hemodynamic stability and enhancing the likelihood of favorable outcomes.

Some of the most commonly used drugs include the following:

Propofol: Preferred for its rapid onset and short duration of action, allowing for quick recovery post-procedure and provides neuroprotection by reducing cerebral metabolic rate and intracranial pressure. While it may cause unwanted intraoperative hypotension, it has minimal impact on cerebral hemodynamics.

Sevoflurane: This volatile anesthetic is valued for its controllability and favorable cardiovascular profile, and it is associated with intracranial vasodilatation. It is unsuitable for use during cesarean sections, particularly because it could cause local environmental pollution for the medical staff.

Midazolam: A benzodiazepine with a relatively short half-life that provides anxiolysis, amnesia, and sedation with a minimal impact on the patient’s hemodynamic.

Fentanyl: Often used in combination with other agents, fentanyl offers potent analgesia and sedation without significant cardiovascular depression. Its drawbacks are primarily associated with opioid-related side effects, particularly respiratory depression, which is of special concern in patients undergoing CS.

Remifentanil: A short-acting opioid preferred for its ultra-rapid elimination rate; its use could be associated with a clinically significant bradycardia.

Dexmedetomidine: Known for its sedative and analgesic properties without significant respiratory depression. A potential drawback is significant bradycardia especially in bolus doses.

In Table 1, we present the anesthetic regimens used in the largest trials evaluating GA versus CS for ischemic stroke thrombectomy [24,25,26,27,28,29,30,31]. 

## 8. General Anesthesia: Should I Choose Endotracheal Intubation or an Extraglottic Device?

The choice between endotracheal intubation or an extraglottic device laryngeal mask airway depends on the patient’s condition, but choosing the right device in ischemic stroke patients can be challenging. On one side, endotracheal intubation provides a secure airway, allows for controlled ventilation, and reduces the risk of aspiration, making it preferred for patients undergoing lengthy or complex procedures or with risk factors for possible inhalation; however, intubation can cause more severe hemodynamic fluctuations than an extraglottic device and may increase intracranial pressure if not performed carefully. On the other side, laryngeal mask airway is easier and quicker to insert, causes less hemodynamic disturbance, and can be useful in short procedures or when intubation is difficult. However, it is a less secure airway compared to intubation and may not be suitable for all patients, particularly those at high risk of aspiration.

The anesthesiologist should evaluate the feasibility and appropriateness of each device on a case-by-case basis, considering that both have their own advantages and disadvantages [32].

## 9. Intraoperative Management

Maintaining a focus on neuroprotection through careful management of cerebral perfusion and metabolic demands is a key principle in the anesthetic management of ischemic stroke. Anesthesiologists should aim to optimize several critical aspects of patient care during the intraoperative period for those affected by stroke; key areas of focus include communication, ensuring proper ventilation, maintaining stable blood pressure, achieving adequate depth of anesthesia, preventing both hyperglycemia and hypoglycemia, and controlling body temperature in order to avoid hypothermia.

### 9.1. Communication

The most important factor during the intraoperative period is communication. Anesthesiologists should maintain close communication with radiology and neurology teams to coordinate care and ensure all team members are aware of the patient’s status and the thrombectomy plan. In our opinion and experience, discussions among these different clinical specialties should not be limited to the acute care setting but should be continuous. Therefore, it is preferable to organize regular updates and multidisciplinary meetings. These meetings help align goals, introduce and explain novel techniques, and set realistic expectations among different clinical profiles. Such collaboration facilitates collective decision making, ultimately promoting the best possible outcomes for each patient.

### 9.2. Ventilation

Carbon dioxide (CO_2_) significantly affects cerebral vessel tone: hypercapnia (elevated CO_2_ levels) causes vasodilation, increasing cerebral blood flow, while hypocapnia (reduced CO_2_ levels) induces vasoconstriction, decreasing cerebral blood flow [33,34]. There is no consensus on the optimal respiratory rate and tidal volume to maintain target PaCO_2_ levels, even if higher tidal volume has been suggested in this class of patients [35]. On the other side, anesthesiologists should remember that these patients are at higher risk of acute lung injury [36] and the optimal tidal volume should be tailored to each patient.

On the other side, oxygen therapy in stroke patients can prevent hypoxia but may cause hyperoxia-related complications, including increased mortality, delayed cerebral ischemia, and oxidative stress [37]. Routine oxygen supplementation does not improve outcomes, and it is possible to recommend to avoid adjunctive oxygen treatment in non-hypoxic patients, leaving this therapy only to patients with a low arterial oxygen level [38,39].

### 9.3. Blood Pressure

Maintaining optimal blood pressure is critical in ischemic stroke management, as both hypo- and hypertension can worsen cerebral ischemia. Preoperatively, blood pressure should be carefully monitored and controlled, targeting a balance that avoids extremes (with an optimal of 140–160 mmHg [40]). The use of intravenous fluids, vasopressors, and inotropes may be necessary to support blood pressure and cerebral perfusion. During surgery, anesthetics and vasoactive agents may be used to maintain blood pressure within an optimal range [41]. Postoperatively, careful monitoring and management continue to be crucial, as fluctuations can significantly impact recovery and outcomes [41].

### 9.4. Depth of Anesthesia

The depth of anesthesia should be monitored whenever feasible [42]. One significant challenge is the potential difficulty in positioning sensors on the head, which could interfere with the radiologist’s work. However, in the authors’ experience, processed EEG sensors rarely interfere with radiologists and typically require only minimal adjustment. Monitoring the depth of anesthesia offers two key benefits. Firstly, it ensures that the patient is adequately sedated. Secondly, it enables a tailored drug approach to ensure neuroprotection, avoiding the risks of oversedation [42,43]. Utilizing a titrated anesthesia plan with short-acting anesthetic agents allows for rapid postoperative neurological assessment, which is critical in stroke patients. Agents such as propofol, remifentanil, and sevoflurane are favored because they permit quick recovery of consciousness, enabling immediate post-procedure evaluation of neurological function.

### 9.5. Blood Glucose and Body Temperature

Controlling blood glucose levels and temperature in stroke patients is crucial for optimizing outcomes and minimizing complications. Hyperglycemia can exacerbate brain injury by increasing the size of the infarct and promoting inflammation, while hypoglycemia can deprive the brain of essential energy, worsening neurological deficits [44,45]. Similarly, maintaining normothermia is essential because hyperthermia can increase metabolic demands, leading to further brain damage, while hypothermia can impair coagulation and increase the risk of infections [46,47]. Thus, precise management of blood glucose levels and temperature helps protect brain tissue, support recovery, and reduce the risk of secondary complications. Monitoring and managing electrolyte imbalances, particularly sodium and potassium levels, can also be important in preventing secondary brain injury [48,49,50]. Strategies to support neuroprotection include maintaining optimal blood pressure to ensure adequate cerebral perfusion, avoiding hyperglycemia and hypoglycemia, and managing body temperature to prevent hyperthermia, which can exacerbate brain injury. Additionally, ensuring adequate oxygenation and ventilation is vital to prevent hypoxia and hypercapnia, both of which can adversely affect cerebral blood flow and increase the risk of further brain injury. Anesthetic agents with neuroprotective properties, such as volatile anesthetics, can help reduce the cerebral metabolic rate and protect brain tissue during ischemic episodes.

In summary, effective anesthetic management of ischemic stroke patients involves a combination of close interdisciplinary communication, advanced monitoring techniques, the use of short-acting anesthetics for rapid recovery, and a vigilant focus on neuroprotective strategies to optimize cerebral perfusion and metabolic demands. These approaches collectively contribute to better patient outcomes and reduce the risk of complications during and after the procedure.

## 10. Postoperative Management

Postoperative management focuses on monitoring and supporting neurological function, managing complications, and optimizing recovery. Patients are usually monitored in an intensive care unit or a specialized stroke unit, where dedicated staff can provide intensive observation and care. Close neurological monitoring is essential, with frequent assessments to detect any changes in cognitive or motor functions [51]. This includes regular neurological assessment and monitoring; moreover, blood pressure and other vital signs should be continuously monitored, with interventions as needed to maintain stability and prevent complications such as hypertension or hypotension. Pain management should be effective yet balanced to avoid over-sedation, which can obscure neurological assessment and delay recovery [52]. Additionally, ensuring adequate oxygenation and fluid balance is crucial for optimizing patient outcomes. Early mobilization and rehabilitation, as discussed below, should also be initiated to promote recovery and reduce the risk of long-term deficits.

The modified Rankin Scale (mRS) is a commonly used scale in patients affected by stroke [53]. This scale is used to provide an evaluation of the degree of disability or dependence in daily activities of stroke survivors, ranging from 0 to 6. A score of 0 indicates no symptoms, 1 no significant disability, 2 slight disability, 3 moderate disability, 4 moderately severe disability, 5 severe disability, and 6 death. The mRS is a largely diffused scale used to assess the effectiveness of stroke treatments and long-term outcomes.

Early mobilization and rehabilitation are critical components of postoperative care [54,55,56], aiding in the recovery of function and prevention of complications such as deep vein thrombosis (complication that could occur in all neurological patients with a prolonged immobilization) [57,58]. Patients may require a multidisciplinary approach, including physical therapy, occupational therapy, and speech therapy, to address the various deficits resulting from the stroke.

## 11. Additional Considerations

Other relevant considerations in the anesthetic management of ischemic stroke include the management of anticoagulation and antiplatelet therapy. Many stroke patients are on these medications, and their management requires a delicate balance between preventing thromboembolic events and minimizing bleeding risk during surgery. Given the above, an accurate planning before, during, and after surgery is of paramount importance. Consultations with other specialists in order to determine the best drugs to be used on a case-by-case basis could be necessary.

Temperature management is another important aspect, as hyperthermia can exacerbate neuronal injury [59]. Maintaining normothermia through active warming or cooling techniques may be necessary. Additionally, glycemic control is critical, as hyperglycemia can worsen ischemic brain injury and a tight glucose control is recommended throughout the perioperative period [60,61,62].

## 12. Novel Promising Technologies

Stroke patients present unique challenges for anesthesiologists, making a tailored anesthesia plan essential for optimal outcomes; for this reason, novel technologies that enhance precision in sedation and hemodynamic monitoring are therefore highly valuable in this context. For instance, advanced sedation monitoring tools like processed electroencephalography (EEG) and bispectral index (BIS) enable real-time tracking of brain activity, allowing anesthesiologists to adjust sedation levels with greater accuracy. These technologies provide continuous feedback on cortical activity, ensuring that patients are not overly sedated, which can minimize recovery time and reduce the risk of sedation-related complications [63]. Excessive sedation, as discussed earlier in relation to drug management, can negatively impact hemodynamics, potentially lowering cerebral perfusion pressure, which is critical for maintaining adequate brain function. Ensuring optimal cerebral perfusion is vital to improving neurological outcomes in stroke patients, and precise sedation management is key to achieving that balance.

In addition to sedation monitoring, innovations in hemodynamic monitoring could play a crucial role in stroke management. In fact, poor cerebral perfusion increases the risk of ischemia or hemorrhagic transformation. Traditional methods of monitoring blood pressure and heart rate may not provide enough detail to manage these risks effectively; however, newer noninvasive or minimally invasive continuous arterial pressure monitoring devices offer dynamic and real-time feedback on systemic and cerebral blood flow [64]. These devices allow clinicians to track cerebral perfusion more accurately, ensuring that blood pressure targets are maintained and cerebral autoregulation is supported during procedures.

Additionally, advanced technologies like near-infrared spectroscopy (NIRS) enable direct monitoring of cerebral oxygenation and blood flow, offering critical data to tailor anesthesia and hemodynamic management [65].

NIRS works by measuring changes in hemoglobin oxygenation in the brain, providing critical insights into cerebral perfusion during stroke interventions. This type of monitoring allows anesthesiologists to monitor regional brain oxygenation continuously, helping to detect early signs of hypoxia or ischemia and enabling prompt adjustments in anesthetic or hemodynamic management. Unlike traditional hemodynamic monitoring, which focuses on systemic blood pressure, NIRS provides direct information about cerebral oxygen delivery, making it a valuable tool in guiding individualized care.

## 13. Future Directions

Future research should focus on optimizing patient outcomes through several key areas. Firstly, further studies, so larger and more diverse clinical trials, are needed to establish clearer guidelines on the comparative efficacy of general anesthesia versus conscious sedation in terms of both immediate procedural success and long-term functional recovery. Secondly, research should move its focus from the CS or GA debate to the choice of anesthetic agent and investigating individualized anesthetic strategies tailored to specific patient profiles, including genetic and physiological factors. Additionally, advancements in real-time monitoring technologies, such as enhanced POCUS and novel biomarkers, may offer more insights into intraoperative brain function and hemodynamic status, potentially guiding more effective anesthetic interventions.

## 14. Conclusions

In conclusion, the anesthetic management of ischemic stroke involves a multifaceted approach that includes preoperative optimization, careful selection of anesthetic agents, strategic airway management, precise blood pressure control, and thorough postoperative care. By adhering to these principles and incorporating specific tips and tricks, anesthesiologists can play a crucial role in optimizing outcomes for patients undergoing surgical treatment for ischemic stroke.

## Figures and Tables

**Figure 1 diagnostics-14-02113-f001:**
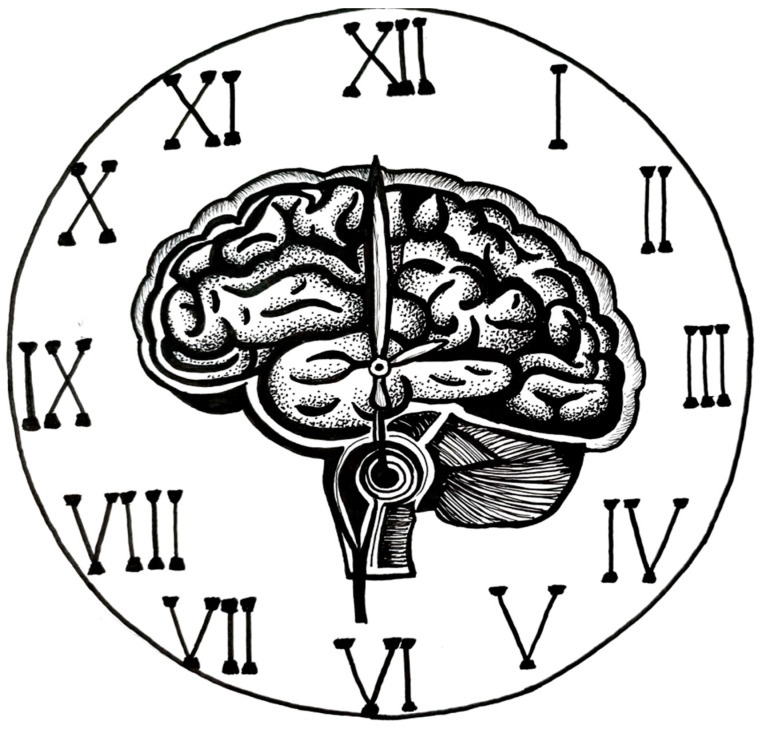
Time is brain.

**Table 1 diagnostics-14-02113-t001:** Anesthetic regimens used in previous randomized controlled trials.

Study (Year)	GA Induction	GA maintenance	CS	Study Main Outcomes
GASS trial (2022) [24]	Etomidate (0.25 to 0.4 mg/kg)	Target-controlled infusion Propofol (maximum target, 4 μg/mL) and Remifentanil (0.5 to 4 ng/mL)	Target-controlled infusion Remifentanil (maximum target, 2 ng/mL)	Functional outcome at three months is similar between CS and GA.
CANVAS II (2022) [25]	Propofol (1 to 2 mg/kg) and Remifentanil (0.2 to 0.8 μg/kg/min).	Propofol and Remifentanil (0.2 to 0.8 μg/kg/min).	Propofol (0.3 to 0.5 mg/kg) and then continuous infusion of Remifentanil (0.01 to 0.06 μg/kg/min) and Propofol (1 to 2 mg/kg/h)	CS was not superior to GA in patients with acute posterior circulation stroke.
AnStroke trial (2017) [26]	Propofol and Remifentanil	Sevoflurane and Remifentanil	Remifentanil infusion	Functional outcome at three months is similar between CS and GA.
SIESTA trial (2016) [27]	Same but higher doses than CS	Same but higher doses than CS	Low-dose short-acting analgesics and, if necessary, sedatives	Functional outcome at 24 h is similar between CS and GA in patients suffering anterior circulation ischemic stroke.
GOLIATH trial (2018) [28]	Alfentanil (bolus 0.02–0.03 mg/kg) and Propofol (bolus 1–5 mg/kg	Propofol 2–10 mg/kg/h) and Remifentanil (0.2–1 µg/kg/min)	Fentanyl 25–50 µg (repeated as necessary); Propofol infusion 1–2 mg/kg/h.	General anesthesia does not result in more infarct growth compared with conscious sedation during endovascular therapy for stroke.
CANVAS trial (2020) [29]	Sufentanil of 0.2 μg/kg and target-controlled infusion with Propofol of 1 to 4 μg/mL	Propofol at 1 to 4 μg/mL and Remifentanil at 0.1 to 0.2 μg/kg/min	Sufentanil 0.1 μg/kg bolus and Propofol 0.5 to 1.0 μg/mL	NIHSS score, rate of successful reperfusion, and functional recovery are similar between CS and GA groups.
Ren et al. (2020) [30]	Propofol 1.5 mg/kg, Fentanyl 2 μg/kg	Propofol 4–6 mg/kg/h, Remifentanil 0.05–0.1 μg/kg, and Dexmedetomidine 0.2–0.4 μg/kg	Propofol (1–1.5 mg/kg; 2–4 mg/kh/h), Dexmedetomidine 0.4–0.7 μg/kg/h, Fentanyl 1 μg/kg, and Midazolam 0.04 mg/kg	Functional outcome and mortality at three months are similar between CS and GA.
Hu et al. (2021) [31]	Alfentanil (0.02–0.03 mg/kg) and Propofol (1–5 mg/kg)	Propofol 2–10 mg/kg/h and Remifentanil (0.2–1 μg/kg/min)	Fentanyl 25–50 µg (repeated as necessary) and Propofol 1–4 mg/kg/h	Functional outcome is similar between CS and GA in posterior circulation stroke; however, CS was associated with bigger final infarct volume.

CS: Conscious sedation; GA: General anesthesia.

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
