# Peer review of "Anesthetic Management of Acute Ischemic Stroke Undergoing Mechanical Thrombectomy: An Overview"

_diagnostics, 2024, doi:10.3390/diagnostics14192113_

Round 1
Reviewer 1 Report
Comments and Suggestions for Authors
The present manuscript entitled “Anesthetic Management of Acute Ischemic Stroke undergoing Mechanical Thrombectomy: an Overview” led by Cassai et al discussed the need of Anesthetic Management of Acute Ischemic Stroke. However, the manuscript needs to be improved.
1. Merge the paragraphs in the abstract.
2. Line 49: add a comma after stroke.
3. Line 51: For this reason, is repeated two times in the same sentence. Modify the sentence.
4. Line 53 and 54: Merge it with the next sentences (line 55).
5. In several places, commas can be inserted throughout the manuscript.
6. Figure 1 is not novel.
7. Lines 84-90: The font is different than the remaining font. Correct it.
8. Line 121-122: must be removed or revised.
9. Line 136: Remove the full stop after exam.
1. Line 188-189: must be removed or revised.
11. Remove — from the lines 207 and 208.
12. Line 225-239: Explain in detail the different anesthetics used while mechanical thrombectomy is required. Clearly explain the advantages and disadvantages of these anastatic agents.
13. The text in the Table 1 font is different than the remaining text. Modify it.
14. Line 244-259: Add more references.
15. Remove the symbol ●
16. The details mentioned in the manuscript are routinely practiced by anaesthesiologists while performing mechanical thrombectomy. What is the novelty of the present manuscript to be emphasized?
17. The limitations and advantages of the present manuscript are to be addressed in the revised manuscript.
18. Several sentences were not connected to the previous sentences or forward sentences. Authors need to carefully revise the entire manuscript.
19. Several punctuation errors (spaces, commas and full stops) to be critically identified and corrected.
210. Keep uniform text font.
Comments on the Quality of English LanguageModerate English editing is required
Author Response
Q1. Merge the paragraphs in the abstract.
A1: Taken
Q2. Line 49: add a comma after stroke.
A2: Taken
Q3. Line 51: For this reason, is repeated two times in the same sentence. Modify the sentence.
A3: Modified in “Consequently”
Q4. Line 53 and 54: Merge it with the next sentences (line 55).
A4:Taken
Q5. In several places, commas can be inserted throughout the manuscript.
A5: We review the manuscript to insert commas wherever needed
Q6. Figure 1 is not novel.
A6:Actually figure 1 has been had-drawn by one of the authors, however similar pictures and the concept of “time is brain” is not novel
Q7. Lines 84-90: The font is different than the remaining font. Correct it.
A7: Corrected
Q8. Line 121-122: must be removed or revised.
A8:Modified as follows: “As stated above the priority of the anesthesiologists is to reduce the compressible times as much as possible in order to reduce the final damage the patient will suffer, however, during this brief window, the anesthesiologist should also assess tests that couldoffer crucial insights into the patient's condition”
Q9. Line 136: Remove the full stop after exam.
A9. Taken
Q10. Line 188-189: must be removed or revised.
A10: Modified as follows:”Given this context, it is evident that the anesthesiologist managing such a frail patient must rely on their own experience, institutional protocols, and the patient's specific conditions to make a decision; however each approach has both advantages and disadvanteges that we will now briefly discuss.”
Q11. Remove — from the lines 207 and 208.
A11:Taken
Q12. Line 225-239: Explain in detail the different anesthetics used while mechanical thrombectomy is required. Clearly explain the advantages and disadvantages of these anastatic agents.
A12: Improved as follows:”Propofol: Preferred for its rapid onset and short duration of action, allowing for quick recovery post-procedure and provides neuroprotection by reducing cerebral metabolic rate and intracranial pressure. While it may cause unwanted intraoperative hypotension, it has minimal impact on cerebral hemodynamics.
Sevoflurane: This volatile anesthetic is valued for its controllability and favorable cardiovascular profile and it is associated with intracranial vasodilatation. It is unsuitable for use during cesarean sections, particularly because it could cause local environmental pollution for the medical staff.
Midazolam: A benzodiazepine with a relatively short half-life that provides anxiolysis, amnesia, and sedation with a minimal impact on patient hemodynamic
Fentanyl: Often used in combination with other agents, fentanyl offers potent analgesia and sedation without significant cardiovascular depression. Its drawbacks are primarily associated with opioid-related side effects, particularly respiratory depression, which is of special concern in patients undergoing CS.
Remifentanil: A short-acting opioid preferred for its ultra-rapid elimination rate, its use could be associated with a clinical significant bradycardia.
Dexmedetomidine: Known for its sedative and analgesic properties without significant respiratory depression. A potential drawback is significant bradycardia especially in bolus doses..
“
Q13. The text in the Table 1 font is different than the remaining text. Modify it.
A13: Taken
Q14. Line 244-259: Add more references.
A14:I would like to include more references; however, the use of SAD in ischemic stroke has been proposed by only a few studies, making it challenging to provide additional evidence. We are open to incorporating further references if the Reviewer could suggest specific articles to be included.
Q15. Remove the symbol ●
A15: Taken
Q16. The details mentioned in the manuscript are routinely practiced by anaesthesiologists while performing mechanical thrombectomy. What is the novelty of the present manuscript to be emphasized?
A16: Thank you for your comment. Our manuscript is intended as a narrative review, aiming to provide practitioners with a comprehensive overview of current practices in anesthesiology during mechanical thrombectomy. While it does not present novel findings, it serves as a valuable resource by summarizing and consolidating existing knowledge on the topic for clinical application.
Q17. The limitations and advantages of the present manuscript are to be addressed in the revised manuscript.
A17: Our manuscript is a narrative review, which typically does not include 'limitations and advantages' in the same way a clinical trial or systematic review would. For comparison, here are some references to other narrative reviews published in the same journal:
https://www.mdpi.com/2075-4418/14/16/1808
https://www.mdpi.com/2075-4418/14/16/1788
https://www.mdpi.com/2075-4418/14/16/1750
Q18. Several sentences were not connected to the previous sentences or forward sentences. Authors need to carefully revise the entire manuscript.
A18:Taken
Q19. Several punctuation errors (spaces, commas and full stops) to be critically identified and corrected.
A19:Taken
Q20. Keep uniform text font.
A20: Taken
Reviewer 2 Report
Comments and Suggestions for Authors
1. Introduction:
1) It does not clearly define the specific research gap or the unique contribution of the current review. The introduction would benefit from a more explicit statement of what existing reviews have missed and what this paper aims to address.
2) The introduction includes broad statements, such as "Anesthetic management plays a crucial role in the surgical treatment of ischemic stroke," without citing specific studies or data to support these claims. Specific references to key studies should be added.
2. Lack of systematic approach:
The Methods section should clarify the approach used to select the studies and literature to review. There is no clear description of the databases searched, keywords used, inclusion/exclusion criteria, or the time period of the literature reviewed. These details should be added.
3. Ambiguity in study selection:
1) The paper mentions various anesthetic strategies but does not specify how studies comparing these strategies were chosen. It would be beneficial to include a flowchart or description of the study selection process to show how the final pool of reviewed studies was determined.
2) The Methods section does not describe how the quality of the included studies was assessed. Include a discussion of the criteria used to assess the methodological rigor of the reviewed studies (e.g., risk of bias, study design).
4. The abstract needs to be revised to conform to the conference format.
5. Please also edit the references in the text to match the format of this journal.
6. Please add more details about Figure 1. And I wonder if Figure 1 is a unique figure.
7. Choosing the right drugs:
1) When performing ischThe Results section presents the results of various studies, but does not consistently compare them in a way that highlights trends or key differences. In this section, add a table summarizing the main results of each study regarding anesthesia management strategies.
2) The review summarizes the results of several studies but lacks quantitative analysis that would provide a more robust assessment of the anesthesia strategies compared. Add an analysis of quantitative results.
8. Lack of future research directions:
The discussion does not clearly identify areas where further research is needed.
Author Response
Q1) It does not clearly define the specific research gap or the unique contribution of the current review. The introduction would benefit from a more explicit statement of what existing reviews have missed and what this paper aims to address.
A1: We tried to better state the aim of this paper by using the following statement:”To guarantee the best outcome for the patient suffering from stroke the anesthesiologist must consider multiple factors when approaching a patient affected by ischemic stroke. Given the complexity of this issue, our narrative review aims to provide practitioners with a comprehensive overview of anesthetic management in acute ischemic stroke. To achieve this, we will briefly explore preoperative evaluation, the choice between general anesthesia and conscious sedation, selection of anesthetic agents, and intraoperative care strategies.”
Q2: The introduction includes broad statements, such as "Anesthetic management plays a crucial role in the surgical treatment of ischemic stroke," without citing specific studies or data to support these claims. Specific references to key studies should be added.
A2: The statement cited by the reviewer was indeed supported by a reference. We are open to adding additional references; however, we kindly ask the reviewer to specify which sentences require further citation for support.
Q3: The Methods section should clarify the approach used to select the studies and literature to review. There is no clear description of the databases searched, keywords used, inclusion/exclusion
Q4: The paper mentions various anesthetic strategies but does not specify how studies comparing these strategies were chosen. It would be beneficial to include a flowchart or description of the study selection process to show how the final pool of reviewed studies was determined.)
Q5:The Methods section does not describe how the quality of the included studies was assessed. Include a discussion of the criteria used to assess the methodological rigor of the reviewed studies (e.g., risk of bias, study design).
Q8 The review summarizes the results of several studies but lacks quantitative analysis that would provide a more robust assessment of the anesthesia strategies compared. Add an analysis of quantitative results.
A3 & A4 & A5&Q8: We would like to clarify that our paper is a narrative review, not a systematic review. Systematic reviews typically involve comprehensive study searches, PRISMA flowcharts, and risk of bias assessments, which are not standard for narrative reviews. For your reference, we are providing examples of recent narrative reviews published in this same journal.
https://www.mdpi.com/2075-4418/14/16/1808
https://www.mdpi.com/2075-4418/14/16/1788
https://www.mdpi.com/2075-4418/14/16/1750
Moreover, quantitative analysis is outside the scope of a narrative review and is more adherent to a systematic review with a meta-analysis
Q6. The abstract needs to be revised to conform to the conference format.
A6: According to the journal guideline the abstract should be untstructured
Q7:. Please also edit the references in the text to match the format of this journal.
A7: We reviewed the references and we believe they match the format of the journal. Please let us know if we make any mistake.
Q6. Please add more details about Figure 1. And I wonder if Figure 1 is a unique figure.
A6:Figure 1 was created by hand by one of the authors. Could you please specify which additional details you think should be included? We believe the figure is self-explanatory in the context of the 'time is brain' concept.
Q7. Choosing the right drugs:
1) When performing ischThe Results section presents the results of various studies, but does not consistently compare them in a way that highlights trends or key differences. In this section, add a table summarizing the main results of each study regarding anesthesia management strategies.
A7 Added a column highlighting study findings
Q9:The discussion does not clearly identify areas where further research is needed.
A9: We added the following paragraph
Future Directions
Future research should focus on optimizing patient outcomes through several key areas. Firstly, further studies, larger and more diverse clinical trials are needed to establish clearer guidelines on the comparative efficacy of general anesthesia versus conscious sedation in terms of both immediate procedural success and long-term functional recovery. Secondly research should move his focus from the CS or GA debate to the choice of anesthetic agent and investigating individualized anesthetic strategies tailored to specific patient profiles, including genetic and physiological factors. Additionally, advancements in real-time monitoring technologies, such as enhanced POCUS and novel biomarkers, may offer more insights into intraoperative brain function and hemodynamic status, potentially guiding more effective anesthetic interventions.
Reviewer 3 Report
Comments and Suggestions for Authors
Thank you for selecting such an interesting topic for review, however I doubt that in its current form it could be a publishable manuscript.
Abstract: you should add the material and method section, as well as the objective of the study.
Introduction: figure 1 is not necessary, it does not provide relevant information, it can be eliminated.
This section is too long, it should be reduced, providing the necessary information to understand and justify this manuscript. It should clearly state the objective of the study, as well as its justification and hypothesis.
Material and method: lacks a section as important as the methodology.
Results: deficient
Discussion: this section does not exist in a clear manner
Author Response
Q1:Thank you for selecting such an interesting topic for review, however I doubt that in its current form it could be a publishable manuscript.
A1:Thank you for your comment. We appreciate the time you spent on our manuscript
Q2:Abstract: you should add the material and method section, as well as the objective of the study.
A2: As per journal requirements the abstract is unstructured. The objective is reported as this:This narrative review highlights key aspects of anesthetic management in ischemic stroke, focusing on preoperative evaluation, anesthetic choices, and intraoperative care
Q3:Introduction: figure 1 is not necessary, it does not provide relevant information, it can be eliminated.
A3: While it is not necessary we believe that this picture vehicolates perfectly our message and for this reason we would like to keep it as it is
Q4:This section is too long, it should be reduced, providing the necessary information to understand and justify this manuscript. It should clearly state the objective of the study, as well as its justification and hypothesis.
A4: We believe that the manuscript has an adequate lenght (please consider that the journal has a requirement of at least 4000 words). We are oped to modify the manuscript under your guidance
Q5:Material and method: lacks a section as important as the methodology.
Results: deficient
Discussion: this section does not exist in a clear manner
A5: This paper is a narrative review, for this reason, methods, results and discussion are something different from a systematic review. Please for your guidance take a look to other papers published in the same journal
https://www.mdpi.com/2075-4418/14/16/1808
https://www.mdpi.com/2075-4418/14/16/1788
https://www.mdpi.com/2075-4418/14/16/1750
Round 2
Reviewer 1 Report
Comments and Suggestions for Authors
Authors have addressed all the issues
Author Response
We would like to thank the reviewer for his comments
Reviewer 2 Report
Comments and Suggestions for Authors
Thank you for your efforts.
Congratulations on completing your paper.
Author Response

(The authors gave the same response as above.)
